# MsSVT: Mixed-scale Sparse Voxel Transformer for 3D Object Detection on Point Clouds

**Shaocong Dong**[1],[*] **Lihe Ding**[1],[*], **Haiyang Wang**[2], **Tingfa Xu**[1],[†],
**Xinli Xu**[1], **Ziyang Bian**[1], **Ying Wang**[1], **Jie Wang**[1], **Jianan Li**[1],[†]
[1]Beijing Institute of Technology, [2]Peking University.
{dsc1010295799, dean.dinglihe, xxlbigbrother, jwang123bit}@gmail.com,
wanghaiyang@stu.pku.edu.cn, {lijianan,ciom_xtf1}@bit.edu.cn

## Abstract

3D object detection from the LiDAR point cloud is fundamental to autonomous driving. Large-scale outdoor scenes usually feature significant variance in instance scales, thus requiring features rich in long-range and fine-grained information to support accurate detection. Recent detectors leverage the power of window-based transformers to model long-range dependencies but tend to blur out fine-grained details. To mitigate this gap, we present a novel Mixed-scale Sparse Voxel Transformer, named MsSVT, which can well capture both types of information simultaneously by the divide-and-conquer philosophy. Specifically, MsSVT explicitly divides attention heads into multiple groups, each in charge of attending to information within a particular range. All groups' output is merged to obtain the final mixed-scale features. Moreover, we provide a novel chessboard sampling strategy to reduce the computational complexity of applying a window-based transformer in 3D voxel space. To improve efficiency, we also implement the voxel sampling and gathering operations sparsely with a hash map. Endowed by the powerful capability and high efficiency of modeling mixed-scale information, our single-stage detector built on top of MsSVT surprisingly outperforms state-of-the-art two-stage detectors on Waymo. Our project page: `https://github.com/dscdyc/MsSVT`.

## 1 Introduction

3D object detection has received increasing attention due to its successful autonomous driving applications. Unlike 2D images with a regular structure of pixels, LiDAR point clouds are naturally irregular and unordered. Hence directly applying CNN-like operations [11, 12] to them can be difficult. To solve this, many researchers have rasterized point clouds into regular voxel grids [24] and employed 3D CNNs to extract high-dimensional voxel features. With the recent rise of vision transformer (ViT) [42] on 2D images, some attempts have been made to generalize global or more efficient window-based transformers to 3D voxels [23] or pillars [7]. These methods successfully seek long-range context by utilizing transformers' powerful abilities in modeling long-range information. However, they ignore that blindly increasing receptive fields would easily blur fine-grained details, especially in sparse 3D space, crucial to accurate object recognition and localization.

Standard window-based transformers update the features of queries in a local window by attending to keys from the same window. Hence simultaneously aggregating long-range context and fine-grained details require enlarging the window size to embrace local and distant voxels. Nevertheless, directly gathering all the voxels within the window as keys suffers a cubical growth of the computational load with the window size. Some attempts alleviate this by sampling only a certain number of key

---

[*]Equal contribution. [†]Correspondence to: Jianan Li and Tingfa Xu.

36th Conference on Neural Information Processing Systems (NeurIPS 2022).

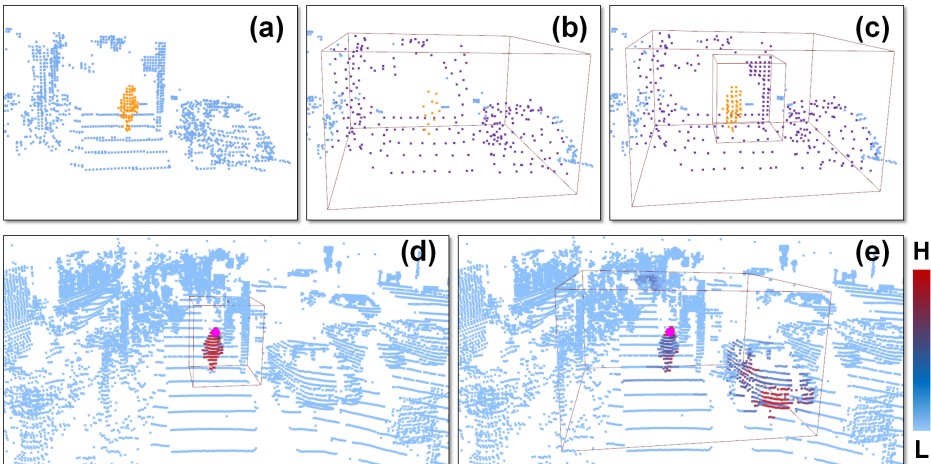

Figure 1: **Top**: In contrast to sampling key voxels from **b)** a single-scale 3D window in **a)** raw point clouds, our MsSVT samples key voxels from **c)** multi-scale windows, thus keeps finer granularity on the target object while covering large-scale neighborhood. **Bottom**: Different head groups accept keys sampled from windows of different scales, and are respectively responsible for obtaining **d)** fine-grained details and **e)** long-range context (reflected by higher attention weights), thus together contribute to accurate object detection collaboratively.

voxels [23]. While a trivial sampling strategy quickly leads to the sparse sampling of local voxels (Fig. 1 b), thus bias mainly on long-range context. To mitigate the above, we try to set up multiple key windows of varying sizes centered on a query window and sample the same number of local and distant key voxels separately from the smaller and larger windows. As a result, we can keep finer granularity in the local region to retain fine-grained details while collecting distant voxels roughly to enlarge the receptive field (Fig. 1 (c)).

With the sampled voxels ready, the next question is how to effectively attend to voxels from different windows and simultaneously capture long-range context and fine-grained details. We argue that the divide-and-conquer philosophy can satisfactorily resolve this issue. Specifically, inspired by the recent findings [42, 59, 26] that transformers learn different levels of self-attention by different heads, we propose a novel Mixed-scale Sparse Voxel Transformer (MsSVT), which explicitly divides the transformer heads into multiple groups. Different head groups accept voxels sampled from windows of different sizes, so they are each in charge of capturing information of a particular scale. Combining the outputs from all the head groups, we can capture mixed-scale information, *i.e.*, long-range context, and fine-grained details. We also design a novel scale-aware relative position encoding strategy to adaptively adjust the position encoding used in each head group according to the range of the keys. We provide some resulting attention maps by two different head groups (Fig. 1 (d), (e)). It is also worth mentioning that the mixed-scale attention enables information exchange across local windows, making MsSVT more compact by saving additional shift window operation commonly required by window-based transformers [21, 7].

Moreover, to improve the efficiency of applying transformers in 3D voxel space, we strive to reduce computational costs in two ways. First, we propose a novel chessboard sampling (CBS) strategy to reduce the number of query voxels that need to be sampled within the query window, to reduce computational costs without losing information. Specifically, we partition the query window into chess-like spaced, and termed as "×", "◯", "△", "□" positions separately. During each attention layer, only one specific position of voxels is sampled and updated by serving as queries, and the updates of the other voxels can be obtained by interpolation. Four positions are selected in the circular. Thus, we can update all the voxels without introducing deviation. Second, we take advantage of non-empty voxels' sparsity by performing mixed-scale window-based attention solely on non-empty sites in 3D space. Furthermore, we parallelize the search and feature gathering for non-empty voxels using hash mapping for further acceleration.

We build a 3D detector by replacing the original sparse 3D CNN backbone in SECOND [50] with our MsSVT and conduct extensive experiments on the large-scale Waymo open dataset [38]. Benefiting from the powerful capability of abstracting mixed-scale voxel features, our single-stage detector based on MsSVT surprisingly outperforms state-of-the-art two-stage detectors. We summarize our contributions as follows:

- We present a novel Mixed-scale Sparse Voxel Transformer (MsSVT), which simultaneously abstracts voxel features with long-range context and fine-grained details.

- We design an efficient chessboard sampling strategy to vastly reduce the computational cost of applying a voxel-based transformer in 3D space and sparsely implement all operations to improve efficiency.

- Our MsSVT-based single-stage detector outperforms state-of-the-art two-stage detectors on Waymo.

## 2    Related work

**3D detection on point clouds.**    The mainstream 3D object detectors are based on voxels [3, 51, 52, 37, 17, 14, 60, 50, 44, 43] or pillars [16, 32]. VoxelNet [60] utilizes PointNet [27, 32] to aggregate features within each voxel and then apply sparse 3D convolution to generate detection results. SECOND [50] investigates improved sparse convolution to further improve speed. Pointpillar [16] converts the point cloud into pillars such that 2D CNNs can be applied to trade off between accuracy and efficiency. Rapoport-Lavie *et al.* [28] introduce the Cylindrical Coordinates to leverage the natural scanning pattern from LiDAR sensors. Chen *et al.* [2] further operate on both Bird Eye View (BEV) and Range View (RV) in the unified Hybrid-Cylindrical-Spherical (HCS) voxel representation. Reconfigurable Voxels [45] improves the local neighbors searching of each voxel using a random walk scheme. Voxel-FPN [15] and Pillar-in-Pillar [41] adopt a multi-scale voxelization strategy to rasterize the input point cloud into multi-sized voxels and incorporate FPN [20] to aggregate multi-resolution feature maps. Two-stage detectors [19, 35, 36, 31] refine the bounding boxes output by a single-stage detector by aggregating raw point clouds or voxel features, yielding state-of-the-art performance.

**Vision transformer.**    Transformer [42, 5] has recently achieved great success in computer vision [6, 1, 21, 18, 61, 53, 46]. Swin-transformer [21] restricts self-attention to non-overlapping local windows while allowing cross-window connection to improve efficiency. SSA [29] divides attention heads into multiple groups to aggregate image features with different granularities. Guo et al. [9] and Zhao et al. [58] make the first step towards introducing the transformer for point cloud analysis. Recently, many approaches [47, 57, 25, 10, 22, 49] apply local self-attention to learn richer 3D feature representation. Our work extends the window-based attention on 3D voxels by introducing scale-aware attention learning equipped with novel sampling strategies for the queries and the keys to improve both accuracy and efficiency.

**Voxel transformer for 3D detection.**    VoTr [23] introduces a voxel-based transformer backbone that performs self-attention on sparse voxels with local and dilated attention mechanisms. Our work improves VoTr by introducing window-based attention and optimizing sparse operation. The recent SST [7] follows a single-stride design and the swin-transformer architecture, which performs well on small objects. Nevertheless, SST is implemented based on pillars. The single window size is not conducive to capturing multi-scale features, resulting in unsatisfactory performance on *Vehicle* when simultaneously detecting multiple categories. In comparison, our MsSVT can capture mixed-scale information to boost the detection of objects of various scales.

## 3    Method

This section first gives details of the MsSVT block, then its efficient sparse implementation, and finally the 3D detector based on MsSVT.

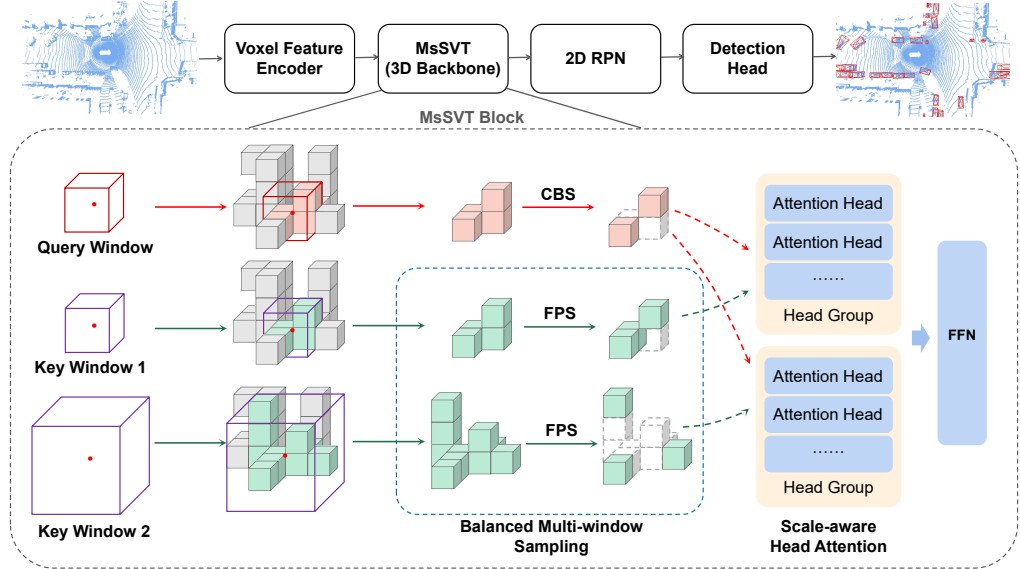

Figure 2: **Top**: overall architecture of our detection network. **Bottom**: details of the MsSVT block. We gather the non-empty voxels within the query window and apply Chessboard Sampling (CBS) to sample the queries. For the keys, we gather the non-empty voxels from the key windows of different sizes separately, and get multiple sets of keys through Balanced Multi-window Sampling, with each set representing information of a specific scale. Keys from windows of different sizes are assigned to different head groups to perform scale-aware attention learning, thus simultaneously capturing both long-range context and fine-grained details.

## 3.1 Mixed-scale Sparse Voxel Transformer

Fig. 2 illustrates the overall architecture of the MsSVT block. We first gather the query and the key voxels via chessboard sampling and balanced multi-window sampling, respectively. The obtained queries and keys are then fed into multiple head groups to capture mixed-scale information through scale-aware attention learning. We further incorporate scale-aware relative position encoding to make better use of position information in different head groups.

### 3.1.1 Balanced Multi-window Sampling

Let $\{\boldsymbol{s}_k|\boldsymbol{s}_k \in \mathbb{Z}^3\}_{k=0}^M$ denote a series of window sizes, where $\boldsymbol{s}_0$ is the size of the query window and $\boldsymbol{s}_{1,...,M}$ are the sizes of $M$ successively larger key windows. Let $\mathcal{V} = \{\boldsymbol{v}_i|\boldsymbol{v}_i = (\boldsymbol{x}_i, \boldsymbol{f}_i)\}_{i=1}^{|\mathcal{V}|}$ be the input voxel set, with $xyz$ coordinates $\boldsymbol{x}_i \in \mathbb{Z}^3$ and feature vector $\boldsymbol{f}_i \in \mathbb{R}^C$ for voxel $i$. We first partition the voxel set into non-overlapping 3D windows each of size $\boldsymbol{s}_0$, and find the non-empty ones as query windows with their centers denoted by $\{\boldsymbol{c}_i|\boldsymbol{c}_i \in \mathbb{Z}^3\}_{i=0}^L$, where $L$ is the total number of query windows. To get query voxels $\mathcal{V}_{\boldsymbol{c}_i,\boldsymbol{s}_0}$ for the query window centered on $\boldsymbol{c}_i$, one can simply gather all the non-empty voxels within the window as the queries. While keeping efficiency in mind, we present a novel chessboard sampling strategy, which will be detailed in Section 3.1.2.

As for the key voxels, instead of sampling within a single large window at once as previous methods [23], which inevitably biases on either local or distant voxels, we simultaneously search for the neighbors for each center $\boldsymbol{c}_i$ within multiple key windows of different sizes. For the key window of size $\boldsymbol{s}_k$, we gather no more than $N_P$ non-empty voxels $\mathcal{V}_{\boldsymbol{c}_i,\boldsymbol{s}_k} = \{\boldsymbol{v}_j| - \boldsymbol{s}_k < \boldsymbol{x}_j - \boldsymbol{c}_i < \boldsymbol{s}_k\}_{j=1}^{N_P}$, where $N_P$ is a pre-set number. Furthermore, to reduce computational cost and keep balanced sampling, we adopt the farthest point sampling (FPS) algorithm to uniformly sample $N_K$ voxels from $\mathcal{V}_{\boldsymbol{c}_i,\boldsymbol{s}_k}$ to obtain the final key voxels $\mathcal{V}_{\boldsymbol{c}_i,\boldsymbol{s}_k}^{fps}, k = 1, ..., M$ at different scales, where $N_K$ is a pre-set maximum number of sampled voxels. Benefitting from the multi-window strategy equipped with uniform sampling by FPS, we can achieve balanced sampling of key voxels at various scales, which is crucial to capturing mixed-scale information.

### 3.1.2  Chessboard Sampling

The prerequisite for us to sample the keys is that we do not have to preserve all the key voxels and only need to select some representative ones to reduce computational redundancy. While things are different for the queries. On the one hand, we should retain and update every query voxel after an attention layer; otherwise, the information could be irreversibly lost. But on the other hand, the computational cost and memory footprint grow dramatically with the window size, making implementation impractical if the queries are not reduced.

Since the positions of non-empty voxels remain unchanged after a series of attention blocks, an effective solution is to sample a subset of query voxels each time for feature updating, and then use the sampled voxels to update those unsampled. Motivated by the above, we present Chessboard Sampling (CBS) as illustrated in Fig. 3. Each voxel in the query window is marked with "×", "◯", "△", or "□", in an evenly spaced pattern. In each MsSVT block, we sample queries from the non-empty voxels with one specific type of the marks, and update the queries' features through attention learning. Then for each unsampled non-empty voxel, we update its feature by finding the K-nearest query voxels (by default $K = 3$) and by linearly interpolating their updated features. The four masks are used in the

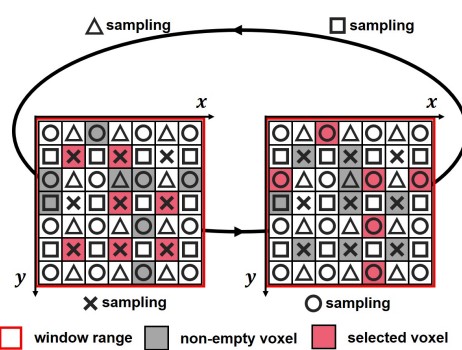

Figure 3: Diagram of chessboard sampling.

circular to sample query voxels in stacked blocks. This way, we retain the original structure and cover all voxels as comprehensively as possible. Note that we can apply the interval sampling on any of the $x$,$y$,$z$ axes to achieve a sampling rate of 1/2, 1/4, or 1/8. Generally, we apply it in the horizontal $x$-$y$ plane (as illustrated in Fig. 3).

### 3.1.3  Scale-aware Head Attention

Given the query voxels $\boldsymbol{\mathcal{V}}_{\boldsymbol{c}_i,\boldsymbol{s}_0} = (\boldsymbol{X}_0, \boldsymbol{F}_0)$ with voxel coordinates $\boldsymbol{X}_0 \in \mathbb{Z}^{N_Q \times 3}$ and feature vectors $\boldsymbol{F}_0 \in \mathbb{R}^{N_Q \times C}$, and the multi-scale key voxels $\boldsymbol{\mathcal{V}}_{\boldsymbol{c}_i,\boldsymbol{s}_k}^{fps} = (\boldsymbol{X}_k, \boldsymbol{F}_k), k = 1, ..., M$ with voxel coordinates $\boldsymbol{X}_k \in \mathbb{Z}^{N_K \times 3}$ and feature vectors $\boldsymbol{F}_k \in \mathbb{R}^{N_K \times C}$, we first get queries $\boldsymbol{Q}$, keys $\{\boldsymbol{K}_k\}_{k=1}^{M}$ and values $\{\boldsymbol{V}_k\}_{k=1}^{M}$ as

$$\boldsymbol{Q},\ \boldsymbol{K}_k,\ \boldsymbol{V}_k = \boldsymbol{F}_0 \boldsymbol{W}^Q,\ \boldsymbol{F}_k \boldsymbol{W}^K,\ \boldsymbol{F}_k \boldsymbol{W}^V, \quad k = 1, ..., M \tag{1}$$

where $\boldsymbol{W}^Q, \boldsymbol{W}^K, \boldsymbol{W}^V \in \mathbb{R}^{C \times C}$ are linear projections. To achieve scare-aware attention learning, we divide multiple attention heads into $M$ groups and assign the keys from windows of different sizes to different head groups. Accordingly, we also split the feature channels of the queries $\boldsymbol{Q}$ into $M$ groups. The $k$-th channel group of $\boldsymbol{Q}$, denoted by $\boldsymbol{Q}_k = \boldsymbol{Q}[:, (k-1) \times C/M : k \times C/M], k = 1, ..., M$, is fed into the $k$-th head group. As a result, each head group is responsible for learning to attend information at a specific scale. The attended feature for the $k$-th head group is

$$\tilde{\boldsymbol{Y}}_{\boldsymbol{k}} = \mathbf{MHA}(\boldsymbol{Q}_k, \boldsymbol{K}_k, \boldsymbol{V}_k, \mathbf{RPE}(\boldsymbol{X}_0, \boldsymbol{X}_k)), \tag{2}$$

where $\mathbf{MHA}(\cdot)$ denotes multi-head-group attention and $\mathbf{RPE}(\cdot)$ represents the newly proposed relative position encoding which will be detailed in Section 3.1.4. Each window size corresponds to a head group with one or more attention heads. We concatenate the output from all heads groups $\{\tilde{\boldsymbol{Y}}_k\}_{k=1}^{M}$ to $\tilde{\boldsymbol{Y}} \in \mathbb{R}^{L \times N_Q \times C}$, which is further fed into a feed-forward network (FFN) implemented by multi-layer perceptron (MLP) to obtain the final mixed-scale feature $\boldsymbol{Y}$:

$$\tilde{\boldsymbol{Y}} = \mathbf{CAT}(\tilde{\boldsymbol{Y}}_1, ..., \tilde{\boldsymbol{Y}}_M), \tag{3}$$

$$\boldsymbol{Y} = \mathbf{MLP}(\mathbf{LN}(\tilde{\boldsymbol{Y}}))) + \tilde{\boldsymbol{Y}}, \tag{4}$$

where $LN(\cdot)$ represents layer normalization.

### 3.1.4 Scale-aware Relative Position Encoding

Relative position encoding is necessary for transformer-based networks because fine-grained position information may be lost in high-level features with the deepening of the network. To make better use of position information to facilitate multi-scale feature learning in our case, we adopt a scale-aware adaptive relative position encoding strategy inspired by [30, 48, 55], which can generate the positional bias dynamically with scales for different head groups.

Specifically, we establish a learnable embedding table $\boldsymbol{T}_k \in \mathbb{R}^{D \times R}$ for the $k$-th head group according to the size of the largest key window [21], where $R$ is the number of possible relative position pairs and $D$ is the feature dimension. We get the relative positional bias for the queries as:

$$\boldsymbol{B}_k^Q = \mathcal{G}(\boldsymbol{Q}_k\boldsymbol{T}_k, \boldsymbol{I}_k) \in \mathbb{R}^{L \times N_Q \times N_K}, \tag{5}$$

where $\boldsymbol{I}_k \in \mathbb{Z}^{L \times N_Q \times N_K}$ represents the table indices corresponding to the actual relative position between the queries and the keys, $\mathcal{G}(\cdot)$ is the operation of gathering features according to the indices. Similarly, we can ge the relative positional bias $\boldsymbol{B}_k^K \in \mathbb{R}^{L \times N_Q \times N_K}$ for the keys. Finally, the biases act directly on the attention weights, and Eq. (2) can be rewritten as:

$$\tilde{\boldsymbol{Y}}_{\boldsymbol{k}} = \sigma\big(\frac{\boldsymbol{Q}_k\boldsymbol{K}_k^\mathsf{T}}{\sqrt{D}} + \boldsymbol{B}_k^Q + \boldsymbol{B}_k^K\big)\boldsymbol{V}_k, \tag{6}$$

where $\sigma(\cdot)$ represents softmax. This way, position embeddings can be adjusted adaptively according to different scales to guide the scale-ware head attention better.

## 3.2 Sparse Implementation

To leverage the natural sparsity of point clouds and further improve efficiency, we sparsely implement all our window center searching, window gathering, and balanced window sampling into CUDA operations. These operations are mainly based on a hash map that establishes the mapping from coordinate space to voxel index [23]. Taking the window gathering operation as an example, we query each possible position w.r.t. the given center within the window, and retrieve the corresponding features if the position is a valid key in the pre-built hash map. More details are available in the supplementary materials.

## 3.3 Detector Establishment

We build our 3D backbone by stacking multiple MsSVT blocks, as shown in Fig. 2. Noted that we set both the query and the key window size in the last MsSVT block as $(1, 1, \infty)$ to compress the 3D voxels into a 2D feature map, where the query is the average voxel features within the pillar window. We replace the 3D backbone in SECOND [50] with MsSVT and keep the other network components unchanged. We discard the down sampling process because MsSVT can already capture features at different scales. The input point cloud is first converted into regular voxels and fed into our MsSVT backbone to get mixed-scale voxel features. The features are then compressed vertically and sent to the subsequent 2D RPN and detection head to get detection results.

# 4 Experiments

In this section, we first provide architectural details of MsSVT, then compare our model with recent state-of-the-art detectors on Waymo Open [38] and KITTI [8] datasets. Thorough ablation studies and in-depth analysis are further provided to validate our design choices.

## 4.1 Architectural Details

MsSVT comprises four regular MsSVT blocks with a query window size of $(3, 3, 5)$ and key window sizes of $(3, 3, 5)$ and $(7, 7, 7)$, followed by a specialized MsSVT block where the windows are set as $1 \times 1$ pillar as mentioned in 3.3. We divide 8 attention heads into 2 head groups. The sampling rate of the chessboard sampling is 1/4 and the maximum number of sampled keys $N_K$ is 32. We use the center head [56] to generate single-stage bounding boxes. In addition, we also provide a two-stage version with CT3D [31]. More details can be found in OpenPCDet [40] since we conduct all experiments with this toolbox.

Table 1: Results on WOD validation set (train with **20**% Waymo data). SS: Single-stage model, TS: Two-stage model, SF: Single frame input. Note that some priors only report results of single-class training, which is generally simpler than multi-class training.

| Method | Reference | Vel_L1 | | Vel_L2 | | Ped_L1 | | Ped_L2 | | Cyc_L1 | | Cyc_L2 | |
|---|---|---|---|---|---|---|---|---|---|---|---|---|---|
| | | mAP | mAPH | mAP | mAPH | mAP | mAPH | mAP | mAPH | mAP | mAPH | mAP | mAPH |
| **Single-Stage Methods** | | | | | | | | | | | | | |
| SECOND [50] | Sensors 2018 | 70.96 | 70.34 | 62.58 | 62.02 | 65.23 | 54.24 | 57.22 | 47.49 | 57.13 | 55.62 | 54.97 | 53.53 |
| PointPillar [16] | CVPR 2019 | 70.43 | 69.83 | 62.18 | 61.64 | 66.21 | 46.32 | 58.18 | 40.64 | 55.26 | 51.75 | 53.18 | 49.80 |
| CenterPoint [56] | CVPR 2021 | 72.76 | 72.23 | 64.91 | 64.42 | 74.19 | 67.96 | 66.03 | 60.34 | 71.04 | 69.79 | 68.49 | 67.28 |
| VOTR-SS [23] | ICCV 2021 | 68.99 | 68.39 | 60.22 | 59.69 | – | – | – | – | – | – | – | – |
| RSN-SF [39] | CVPR 2021 | 75.10 | 74.60 | 66.00 | 65.50 | – | – | – | – | – | – | – | – |
| MsSVT-SS (ours) | – | **77.18** | **76.67** | **68.75** | **68.28** | **80.25** | **73.05** | **72.88** | **66.14** | **73.75** | **72.53** | **70.96** | **69.79** |
| **Two-Stage Methods** | | | | | | | | | | | | | |
| Part-A2 [34] | TPAMI 2020 | 74.66 | 74.12 | 65.82 | 65.32 | 71.71 | 62.24 | 62.46 | 54.06 | 66.53 | 65.18 | 64.05 | 62.75 |
| PV-RCNN [35] | CVPR 2020 | 75.95 | 75.43 | 68.02 | 67.54 | 75.94 | 69.40 | 67.66 | 61.62 | 70.18 | 68.98 | 67.73 | 66.57 |
| Voxel-RCNN [4] | AAAI 2021 | 76.13 | 75.66 | 68.18 | 67.74 | 78.20 | 71.98 | 69.29 | 63.59 | 70.75 | 69.68 | 68.25 | 67.21 |
| PV-RCNN++ [36] | ARXIV 2021 | 77.61 | 77.14 | 69.18 | 68.75 | 79.42 | 73.31 | 70.88 | 65.21 | 72.50 | 71.39 | 69.84 | 68.77 |
| VOTR-TS [23] | ICCV 2021 | 74.95 | 74.25 | 65.91 | 65.29 | – | – | – | – | – | – | – | – |
| CT3D [31] | ICCV 2021 | 76.30 | – | 69.04 | – | – | – | – | – | – | – | – | – |
| MsSVT-TS (ours) | – | **78.41** | **77.91** | **69.74** | **69.17** | **82.34** | **76.77** | **74.71** | **69.36** | **75.74** | **74.65** | **73.72** | **72.64** |

Table 2: Results on WOD validation set (train with **100**% Waymo data).

| Method | Reference | Vel_L1 | | Vel_L2 | | Ped_L1 | | Ped_L2 | | Cyc_L1 | | Cyc_L2 | |
|---|---|---|---|---|---|---|---|---|---|---|---|---|---|
| | | mAP | mAPH | mAP | mAPH | mAP | mAPH | mAP | mAPH | mAP | mAPH | mAP | mAPH |
| **Single-Stage Methods** | | | | | | | | | | | | | |
| SECOND [50] | Sensors 2018 | 72.27 | 71.69 | 63.85 | 63.33 | 68.70 | 58.18 | 60.72 | 51.31 | 60.62 | 59.28 | 58.34 | 57.05 |
| PointPillar [16] | CVPR 2019 | 71.57 | 70.99 | 63.06 | 62.54 | 70.61 | 56.70 | 62.85 | 50.24 | 64.36 | 62.27 | 61.95 | 59.93 |
| SST-SS-SF [7] | CVPR 2022 | 73.57 | – | 64.80 | – | 80.01 | – | 71.66 | – | 70.72 | – | 68.01 | – |
| MsSVT-SS (ours) | – | **77.83** | **77.32** | **69.53** | **69.06** | **80.39** | **73.61** | **73.00** | **66.65** | **75.17** | **73.99** | **72.37** | **71.24** |
| **Two-Stage Methods** | | | | | | | | | | | | | |
| Part-A2 [34] | TPAMI 2020 | 77.05 | 76.51 | 68.47 | 67.97 | 75.24 | 66.87 | 66.18 | 58.62 | 68.60 | 67.36 | 66.13 | 64.93 |
| PV-RCNN [35] | CVPR 2020 | 78.00 | 77.50 | 69.43 | 68.98 | 79.21 | 73.03 | 70.42 | 64.72 | 71.46 | 70.27 | 68.95 | 67.79 |
| PV-RCNN++ [36] | ARXIV 2021 | 79.25 | 78.78 | 70.61 | 70.18 | 81.83 | 76.28 | 73.17 | 68.00 | 73.72 | 72.66 | 71.21 | 70.19 |
| MsSVT-TS (ours) | – | **79.35** | **78.86** | **70.65** | **70.23** | **82.41** | **77.04** | **74.74** | **69.57** | **77.12** | **76.01** | **74.98** | **74.07** |

## 4.2 Results on Waymo

**Setups.** We first evaluate our model on large-scale Waymo Open Dataset [38]. The input is a single-frame point cloud with 150m×150m detection range. We set the detection range in the horizontal and the vertical direction as $[-75.2m, 75.2m]$ and $[-2.0m, 4.0m]$, respectively. The voxel size is $(0.4m, 0.4m, 0.6m)$. We follow the same training strategy as in [23]. Specifically, we train our model for 80 epochs on 20% Waymo data using the Adam optimizer [13]. We apply the cyclic decay scheme [50], by which the learning rate is increased from 1e-4 to 1e-3 during the first 40% epochs and further decreased to 1e-5 in the remaining epochs. We also report the results of training for 30 epochs on 100% Waymo data using the same optimizer and learning rate scheme. The evaluation metric is the 3D mean Average Precision (mAP) for difficulty levels of LEVEL 1 and LEVEL 2.

**Main results.** We compare our model with state-of-the-art priors in Table 1 and Table 2 . It is worth noting that our model performs simultaneous detection of three object categories, which is more challenging than detecting a single category. Using 20% training data, our single-stage MsSVT-SS significantly surpasses other single-stage counterparts, even though some of them are trained specially for one specific category. Noticeably, MsSVT-SS performs on par with the state-of-the-art two-stage PV-RCNN++ [36] on *Vehicle*, and even better (0.8-1.5 mAP higher) on *Pedestrian* and *Cyclist*. In addition, our two-stage MsSVT-TS performs the best on all categories, and significantly outperforms previous best-performing PV-RCNN++ by **2.9** and **3.2** mAP on *Pedestrian* and *Cyclist*, respectively.

Similarly, when using 100% data, both our single-stage MsSVT-SS and two-stage MsSVT-TS achieve the best results compared to their corresponding counterparts. Noticeably, our MsSVT-SS surpasses the recent transformer-based detector SST [7] dedicated to small object detection by large margins of **4.3** and **4.5** mAP on *Vehicle* and *Cyclist*, respectively. This clearly evidences the superiority of our MsSVT in capturing mixed-scale information over conventional transformer designs.

**Visualization.** We provide some qualitative results in Fig. 4. Our MsSVT can still predict accurate bounding boxes in scenes beyond the range of 50m where the points become extremely sparse (Fig. 4 (a)). This evidences that MsSVT successfully captures valuable contextual information in the absence of fine-grained details for long-range object detection. In addition, our model also performs well in complex scenes in the presence of dense objects with large-scale variations (Fig. 4 (b). It should be owed to the flexibility and robustness of MsSVT.

**Analysis.** We further visualize some attention maps which indicate what has been focused on by our MsSVT to understand its behavior better. As shown in Fig. 5, the head group with a smaller key window focuses on the local foreground information, while the head group with a larger key window attends more to the longer-range context. As a result, the two head groups complement each other, well capturing mixed-scale information to support the detection of objects of various scales.

Table 3: Results on KITTI validation set.

| Method | Reference | 3D Car (IoU=0.7) | | | 3D Ped. (IoU=0.5) | | | 3D Cyc. (IoU=0.5) | | |
|---|---|---|---|---|---|---|---|---|---|---|
| | | Easy | Mod | Hard | Easy | Mod. | Hard | Easy | Mod. | Hard |
| **Single-Stage Methods** | | | | | | | | | | |
| SECOND [50] | Sensors 2018 | 86.46 | 77.28 | 74.65 | 61.63 | 56.27 | 52.60 | 80.10 | 62.69 | 59.71 |
| PointPillar [16] | CVPR 2019 | 88.61 | 78.62 | 77.22 | 56.55 | 52.98 | 47.73 | 80.59 | 67.16 | 63.11 |
| 3DSSD [54] | CVPR 2020 | 88.55 | 78.45 | 77.30 | 58.18 | 54.32 | 49.56 | 86.25 | 70.49 | 65.32 |
| VoTr-SS [23] | ICCV 2021 | 87.86 | 78.27 | 76.93 | – | – | – | – | – | – |
| MsSVT-SS (ours) | – | **89.08** | **78.75** | **77.35** | **63.59** | **57.33** | **53.12** | **88.57** | **71.70** | **66.29** |
| **Two-Stage Methods** | | | | | | | | | | |
| PointRCNN [33] | CVPR 2019 | 89.03 | 78.78 | 77.86 | 62.50 | 55.18 | 50.15 | 87.49 | 72.55 | 66.01 |
| Part-A2 [34] | TPAMI 2020 | 88.48 | 78.96 | 78.36 | **70.73** | **64.13** | **57.45** | 88.18 | 73.35 | **70.75** |
| PV-RCNN [35] | CVPR 2020 | **89.35** | 83.69 | 78.70 | 63.12 | 54.84 | 51.78 | 86.06 | 69.48 | 64.50 |
| VoTr-TS [23] | ICCV 2021 | 89.04 | 84.04 | 78.68 | – | – | – | – | – | – |
| MsSVT-TS (ours) | – | 89.32 | **84.66** | **78.94** | 66.11 | 58.94 | 53.86 | **92.49** | **73.60** | 69.34 |

## 4.3 Results on KITTI

**Setups.** We also evaluate our model on KITTI [8]. Given an input point cloud, we reserve the points within the range of $[0\text{m}, 70.4\text{m}]$, $[-40.0\text{m}, 40.0\text{m}]$ and $[-3.0\text{m}, 1.0\text{m}]$ on the $x$,$y$,$z$ axes, respectively. The voxel size is set to $(0.32\text{m}, 0.32\text{m}, 0.4\text{m})$ and other settings remain the same as in the experiments on Waymo. We train the model for 100 epochs with the Adam optimizer. The learning rate is 0.003, decayed by the cyclic scheme [50]. The evaluation metric is 3D mAP for three difficulty levels (easy, moderate, and hard).

**Main results.** Table 3 depicts that our model achieves competitive performance on all three categories. Specifically, on *Car*, our single-stage MsSVT-SS surpasses the superior VoTr [23] by 0.5 mAP. Meanwhile, on *Pedestrian* and *Cyclist*, MsSVT-SS even outperforms some preeminent two-stage detectors. Moreover, our two-stage MsSVT-TS further increases the lead. These results demonstrate that our MsSVT can be well generalized to various datasets.

## 4.4 Ablation Study

All ablation models are trained for 12 epochs on 20% Waymo data. More implementation details and ablations on hyperparameters are available in supplementary materials.

**Balanced multi-window sampling.** We first validate our balanced multi-window sampling strategy in Table 4. The base model listed in the first row employs a 3D version of standard window-based attention [21] with window size $(3, 3, 5)$ and without shift window scheme, adopts dilated key sampling [23], and gathers all non-empty voxels within the window as queries. We build a model variant by simply replacing the dilated key sampling with our proposed balanced multi-window sampling. Results in the second row demonstrate that sampling key voxels from multiple

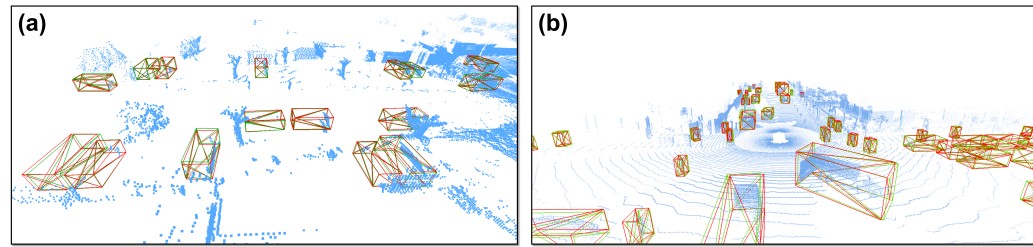

Figure 4: Qualitative results on Waymo. The red and green boxes denote ground-truths and predictions, respectively. Our MsSVT performs impressively in scenes (a) beyond the range of $50$m, and (b) in the presence of dense objects with large scale variations.

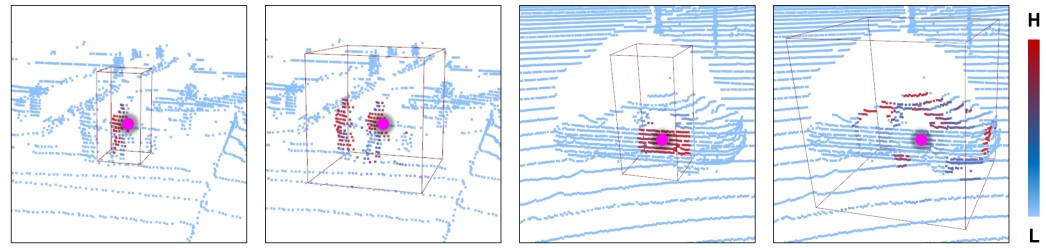

Figure 5: Visualization of attention maps. Pink dot denotes the query position. Positions with high and low attention weights are in red and blue, respectively.

windows of different sizes brings noticeable performance gains, $i.e.$, $71.66 \rightarrow 73.44$ on *Pedestrian* and $63.31 \rightarrow 66.88$ on *Cyclist*.

**Scale-aware head attention.** We also validate the effectiveness of our proposed scale-aware head attention in Table 4. By comparing the results in the second and the third rows, one can find that enabling multiple head groups to capture information at different scales clearly boots the performance.

**Scale-aware relative position encoding.** Table 4 shows that incorporating scale-aware relative position encoding (listed in the last row) further improves the per-

Table 4: Ablations on different components of MsSVT. BMS: Balanced Multi-window Sampling, SHA: Scale-aware Head Attention, SRPE: Relative Position Encoding.

| BMS | SHA | SRPE | Veh / Ped / Cyc |
|:---:|:---:|:---:|:---:|
| | | | $69.51/71.66/63.31$ |
| ✓ | | | $71.24/73.44/66.88$ |
| ✓ | ✓ | | $71.96/75.08/67.16$ |
| ✓ | ✓ | ✓ | $72.37/75.99/67.90$ |

formance compared to the model variant that adopts scale-agnostic position encoding. It well supports our design motivation that the position encoding should vary with different scales.

**Chessboard sampling.** Table 5 reports the performance of the model variants that apply chessboard sampling with different sampling rates and of the model without performing any sampling. We measure the latency on Tesla-V100 GPU and Unbuntu-16.04, Python 3.7, Cuda-10.2, and Pytorch-1.8. Our model is not sensitive to varying sampling rates. Incorporating chessboard sampling with a sampling

Table 5: Ablations on sampling strategy.

| Strategy | Veh / Ped / Cyc | Mem (G) | Lat (ms) |
|:---:|:---:|:---:|:---:|
| w/o | $72.58/75.74/68.24$ | 18.3 | 167 |
| 1/2 | $72.44/76.03/67.81$ | 14.5 | 138 |
| 1/4 | $72.37/75.99/67.90$ | 12.2 | 121 |
| 1/8 | $72.01/75.54/67.43$ | 11.4 | 113 |

rate of 1/4 results in only negligible performance degradation compared to the model variant without sampling, yet largely improves the computational efficiency by a significant reduction in memory footprint (up to $33\%$ decrease) and in latency (up to $28\%$ decrease).

## 5  Conclusion

This paper presents MsSVT, a novel sparse voxel transformer that is capable of capturing mixed-scale information to boost 3D object detection. We begin with sampling the queries and the keys at different scales through the novel chessboard and the balanced multi-window sampling strategy, respectively. Then, we explicitly divide attention heads into multiple scale-aware groups, each in charge of capturing information at a specific scale. Extensive experiments have shown that our MsSVT is superior in detecting objects of various scales and granularities thanks to the learned mixed-scale information.

**Limitations.** MsSVT can capture mixed-scale information from multiple local windows and achieves promising performance. Yet, the sizes of the windows need to be pre-set manually. We will explore the adaptive-window version of MsSVT in future work.

## Acknowledgements

This work was financially supported by the National Natural Science Foundation of China (No. 62101032), the Postdoctoral Science Foundation of China (Nos. 2021M690015, 2022T150050), and Beijing Institute of Technology Research Fund Program for Young Scholars (No. 3040011182111).

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
