# OpenReview forum: "MsSVT: Mixed-scale Sparse Voxel Transformer for 3D Object Detection on Point Clouds"
_NeurIPS.cc/2022/Conference — NeurIPS 2022 Accept_

### Official Review · Reviewer_G2ui · 2022-07-10

**Rating:** 7
**Confidence:** 5
**Soundness:** 3 good
**Presentation:** 3 good
**Contribution:** 3 good

**Summary:**

This paper proposes a new transformer architecture for Lidar-based 3D object detection. Previous windows-based transformers utilize large windows to capture the long-range context information but ignore the negative effect of blurring fine-grained details. This is especially true for 3D point cloud cases where the point downsampling is performed and sparse and distant points are selected (further blurring fine-grained details). To alleviate this, the authors propose to design a multi-scale transformer architecture and set up key windows of different scales in which different windows contain the same amount of downsampled points. Scale-aware head attention is also proposed to divide the attention head into multiple groups and aggregate after inner group computation with the query. Additionally, the authors further propose a chessboard sampling strategy to iteratively sample different query voxels in order to reduce the computational cost while maintaining similar accuracy performance. In the end, the final models are evaluated on the KITTI and Waymo datasets and achieve SOTA results.

**Questions:**

Q1: On line 180, the authors mentioned that the model selected one of the four characters as the sampling location. How are these chessboards generated? Is it purely random? The authors need to provide more details.

Q2: Please address the weakness I listed above.

**Limitations:**

Partially. More discussion of the potential negative societal impact like the earlier adoption of not-yet validated autonomous vehicles (or misleading 3D detection performance) will be useful.

**Strengths And Weaknesses:**

S1: The overall design is sound and relatively novel. This work nicely extends the Single-Stride Transformer [1] to work with voxel data. Concretely, SST only works with pillar (a long elongated voxel) representation to maintain efficiency. Extending windows-based transformers to the voxel representations come with an efficiency challenge due to the larger amount of voxels inside a key window. The authors nicely address this by proposing multi-scale window attention and chessboard query sampling strategy. The overall approach significantly improves the SST baseline and sheds light on efficient transformer design with 3D point cloud data.

S2: The approach is well ablated on two popular datasets and achieves SOTA results compared to published methods.

S3: The paper is generally well written and easy to follow.

W1: There is no ablation on the number of groups for multi-scale attention (currently set to two). To further verify the motivation, the authors need to investigate the accuracy/speed tradeoff with a larger number of groups.

W2: Current model only works with single-frame data. Could it scale to multi-frame?

W3: Experiments with more two-stage architecture will be interesting (minor, this could be done later).

---

> ### Author Response · Authors · 2022-08-02
> **Official Responses of Reviewer G2ui**
>
> We sincerely thank the reviewer for providing thoughtful review and positive feedback. Below are our responses to the questions and suggestions raised by the reviewer.
>
> **R3-Q1: More experiments of the number of groups for multi-scale attention.**
> **R3-A1:** As shown below, we conduct a series of ablative experiments by increasing or decreasing the number of key windows and attention head groups allocated to each window. All models are trained for 12 epochs on 20\% Waymo data. Compared with two windows (M1), more windows (M2, M3, M4) do not improve performance but increase the runtime latency. In our Scale-aware Head Attention, we distribute all feature channels and attention heads among the key windows to make different head groups in charge of a particular scale. Thus too scattered key windows will lead to fewer channels for each head group, which can weaken the representation ability of the single-scale features. Meanwhile, a single window (M5, M6) lacks multi-scale information and leads to low accuracy. Moreover, (M2, M3, and M4) also demonstrate that allocating more heads to the oversized windows significantly affects the accuracy of small objects. These experiments prove that our current combination (M1) can bring superiority of the mixed-scale window into full play and make a trade-off between speed and accuracy.
>
> | Strategy | Comb      | Num Heads | Veh / Ped / Cyc     | Lat (ms) |
> |----------|-----------|-----------|:-------------------:|:--------:|
> | M1*      | (3,7)     | (4,4)     | $72.37/75.99/67.90$ | 121      |
> | M2       | (3,5,7)   | (2,3,3)   | $71.94/75.76/67.07$ | 137      |
> | M3       | (3,5,7)   | (2,2,4)   | $72.17/75.45/67.01$ | 139      |
> | M4       | (3,5,7,9) | (2,2,2,2) | $72.04/75.38/66.53$ | 157      |
> | M5       | (3)       | (8)       | $71.46/74.39/65.55$ | 98       |
> | M6       | (7)       | (8)       | $72.06/75.20/66.33$ | 112      |
>
> **R3-Q2: Could the model scale to multi-frame?**
> **R3-A2:** Yes, our model can easily scale to multi-frame. We scale the model to multiple frames following SST. Due to limited time, we trained 12 epochs on 20\% Waymo. As shown below, the performance of our multi-frame model is much better than that of the single frame, which shows the great potential of our method. In fact, MsSVT is compatible with multi-frame and other tricks since the input and output of our 3D backbone network are normal voxels.
>
> | Frame |   Veh / Ped / Cyc   |
> |:-----:|:-------------------:|
> | 1f    | $72.37/75.99/67.90$ |
> | 3f    | $75.78/79.96/72.44$ |
>
> **R3-Q3: Experiments with more two-stage architectures will be interesting.**
> **R3-A3:** Thank you for your advice. We add the result of LidarRCNN as shown below. Furthermore, we will test more two-stage methods in the future. The performance of MsSVT is much better than the LidarRCNN baseline (about 5-7 mAP) and is comparable to MsSVT-CT3D. It proves that our method can be widely applicable to different architectures.
>
> |Method\\mAP | Vel\_L1| Vel\_L2|Ped\_L1 |Ped\_L2 |Cyc\_L1 | Cyc\_L2|
> |:------:|:------:|:------:|:------:|:------:|:------:|:------:|
> |MsSVT-SS|  77.18 |  68.75 | 80.25  | 72.88  | 73.75  | 70.96  |
> | CT3D   | 76.30  | 69.04  | --     |  --    | --     | --     |
> | MsSVT-CT3D | 78.41 | 69.74 | **82.34**  | **74.71** | **75.74**  | **73.72**  |
> | LidarRCNN(2x) | 73.5 | 64.7 | 71.2   | 63.1   | 68.6   | 66.1   |
> | MsSVT-LidarRCNN | **78.62** | **69.87** | 82.26 | 74.68  | 75.58 | 73.57  |
>
>
> **R3-Q4: More details of chessboard sampling.**
> **R3-A4:** Chessboard sampling is essentially an equal interval sampling strategy. First, we divide the entire voxel mesh space into four characters alternately without considering sparsity. Fig.3 is an example of division on a 2D plane. The "cross" character represents the position where the X coordinate is odd, and the Y coordinate is odd (odd-odd), and "circle", "triangle", and "square" represent even-even, odd-even, and even-odd, respectively. The similar combinations of odd and even can also be extended to three dimensions to get a sampling rate of power-of-2 (we apply it on two dimensions like Fig.3 and do not sample on the Z-axis). After that, we select the "cross" voxels in the first attention layer, which means all the non-empty voxels in the "cross" position will be fetched and fed to the attention layer to serve as queries. In the second attention layer, we choose "circle". In the third, we choose "triangle". And so on. Chessboard sampling has high efficiency and can significantly reduce memory. Moreover, all non-empty positions can be preserved completely after multiple stacked layers, reducing the uncertainty of updating query voxels.
>
> **R3-Q5: More discussion of the potential negative societal impact will be useful.**
> **R3-A5:** Thanks. The misleading 3D detection will result in disastrous consequences such as car crashes. We should continue improving the reliability and robustness of 3D detection.

---

> > ### Comment · Reviewer_G2ui · 2022-08-07
> > **Reply to Response**
> >
> > I thank the authors for the detailed reply. I will keep my original rating.

---

### Official Review · Reviewer_aXhD · 2022-07-11

**Rating:** 5
**Confidence:** 4
**Soundness:** 3 good
**Presentation:** 2 fair
**Contribution:** 2 fair

**Summary:**

This paper proposes a backbone, dubbed MsSVT, for exploiting mixed-scale sparse voxel transformer for LiDAR-based 3D detection. It devises chessboard sampling and balanced multi-window sampling strategies to obtain queries and a set of keys from different scales and then uses multiple attention heads for different scale-aware groups. The mixed-scale features are aggregated for multi-scale scene understanding. Experiments demonstrate the efficacy of the proposed method and the qualitative analysis with expected attention weights shows the role of the mixed-scale design.

**Questions:**

- Does the implementation only try the mentioned two-scale case? How about the results for other alternative designs? For example, using the same sampling strategy for both queries and keys, and using more scales for mixing?


**Limitations:**

The author analyzes the limitations and societal impacts adequately.

**Strengths And Weaknesses:**

Strengths:
- The motivation for designing mixed-scale voxel feature extraction is reasonable.
- The proposed method not only shows impressive performance compared to other methods but also turns out to be efficient with the customized sampling designs. It is important for the practical use of LiDAR-based 3D detectors.
- The illustration, such as Fig. 2 and 5, clearly show the design methods and the functionality of mixed-scale voxelization feature extraction.
- It shows great potential for improving the detection performance of small objects, such as pedestrians and cyclists, in the Waymo experiments.

Weaknesses:
- The motivation for the necessity of mixed-scale voxelization is a little inaccurate.

Large-scale outdoor scenes do not feature significant variance in instance scales, from my perspective, at least compared to 2D object detection in images. A more reasonable explanation should be related to the ununiform distribution of LiDAR point clouds? Due to the sparsity and irregularity of LiDAR points distribution, a voxel-based 3D detector needs customized designs for voxelization to tackle the difficulty of detecting small objects and far-away objects. Of course, I think the scale variance of different instances is also a problem in 3D although it is just one reason for the necessity of mixed-scale voxelization.

Here also needs a literature review for different voxelization methods in previous works, for example:

Every View Counts: Cross-View Consistency in 3D Object Detection with Hybrid-Cylindrical-Spherical Voxelization, NeurIPS 2020

Reconfigurable Voxels: A New Representation for LiDAR-Based Point Clouds, CoRL 2020

It's All Around You: Range-Guided Cylindrical Network for 3D Object Detection

Context-Aware Dynamic Feature Extraction for 3D Object Detection in Point Clouds (Pillar-in-Pillar)

Voxel-FPN: Multi-Scale Voxel Feature Aggregation for 3D Object Detection from LIDAR Point Clouds, Sensors 2020

- Technical contribution: This paper contributes more engineering techniques, including mixed-scale voxelization, multi-scale heads, and efficient sampling implementation. From the perspective of novel technical contributions and new insights to the community, the contribution is just above the borderline.

- Experiments: There are only results on the validation set both on Waymo and KITTI. Although the results on the validation set of Waymo are commonly used for comparison, the validation results on the KITTI are usually not convincing enough. Additional results on the test set would strengthen the experiment part.

- Minor typos: The paper needs polishing and fixing some minor typos, such as line 123 prsent -> present, line 125 regrard -> regard, line 197 wrt -> w.r.t or with respect to. The methodology part can be also improved. For example, the two sections about sampling are not put together, which makes the reader confused about their relationship and the main philosophy of different sampling methods for queries and keys (It is clarified until the beginning of Sec. 3.2).

---

> ### Author Response · Authors · 2022-08-02
> **Official Responses of Reviewer aXhD - Third Part**
>
> **R2-Q6: Difference with the mentioned methods.**
> **R2-A6:** We are very different from the mentioned methods. We will first briefly revisit them and then elaborate on our differences. In addition, all the mentioned works have been involved in the attached revision.
>
> 1) **Every View Counts** replaces the commonly used cuboid-shaped voxel representation with Hybrid-Cylindrical-Spherical (HCS) Voxels by changing $R=\sqrt{x^2 + y^2 + z^2}$ in the Spherical coordinates system into $r=\sqrt{x^2 + y^2}$ which is adopted in Cylindrical coordinate system, then performs detection on both Bird Eye View (BEV) and Range View (RV) in the unified voxel representation. In comparison, **MsSVT** uses the cuboid-shaped voxel and performs detection on BEV. We focus on how to gather voxels and apply an efficient attention mechanism in multiple windows rather than on designing the voxel representation.
>
> 2) **Reconfigurable Voxels** changes the local neighbor searching of each voxel from finding its nearest four neighbors into a random walk scheme, which gets four gathered voxel sets connected to the center voxel as the neighbors through a complex pre-defined rule without learning parameters. In comparison, **MsSVT** explicitly divides the local neighbors into large and small regions by a more efficient and straightforward distance partition. Then the voxels learn to attend to their neighbors with more flexibility, thus leading to more powerful feature extraction. In addition, **MsSVT** models the relationship in local voxel sets where all the voxels in the window, even not connected, can be attended to rather than reconfiguring the neighbors for a single voxel where only connected voxels are considered.
>
> 3) **It's All Around You** introduces the Cylindrical Coordinates to 3D detection to leverage the natural scanning pattern from LiDAR sensors and propose a range-guided convolution to address the problem that similar objects appear across different bins depending on the range when adopting Cylindrical Coordinates. While **MsSVT** follows the traditional Cartesian Coordinates and divides the scene into normal cells, we conquer the sparsity and irregularity of voxels by mixed-scale window transformers. The motivation and solution are very different.
>
> 4) **Voxel-FPN** adopts a multi-scale voxelization strategy to rasterize the input point clouds into voxels with different voxel sizes and utilizes an FPN network to aggregate the multi-resolution feature maps. **Pillar-in-Pillar** further solves the misalignment problem in multi-scale voxelization by leveraging a center-aligned voxelization strategy with overlapped sub-voxel partition.
> Our **MsSVT** is very different from these multi-scale voxelization methods.
>    * **We gather mixed-scale information from different windows rather than apply multiple voxelizations.** First of all, our mixed-scale windows are different from the multi-resolution voxels in two aspects: i) The basic unit of our windows are still high-resolution voxels, while Voxel-FPN pools all the information in a significant voxel, in which case the local structure will be blurred irreversibly. For example, the low-resolution voxelization will convert an object into a solid cube, while our large window roughly collects high-resolution voxels on its surface and thus can still retain the local structure thanks to the well-designed sampling strategy. ii) Our windows are much larger than the multi-resolution voxels (e.g., we have two window sizes (1.2m, 2.8m) while Voxel-FPN adopts three voxel sizes (0.16m, 0.32m, 0.64m)). This way, our MsSVT can obtain sufficient receptive benefit from the large window more directly and efficiently.
>    * **MsSVT can capture mixed-scale information in a single layer while Voxel-FPN relies on a complicated architecture.** Previous multi-scale voxelization methods need first to adopt multiple voxelizations, then stack 3D CNNs for different resolutions to extract multi-scale feature maps, and finally utilize the FPN network to fuse the pyramid feature maps and obtain the multi-scale feature map. Technically, the multiple voxelizations will bring disastrous computational costs to the sparse voxel operation. While our **MsSVT** only voxelizes the point clouds at once at the beginning, then maintains the resolution without down-sampling as SST[7]. We can capture multi-scale information in a single layer by making voxels learn to attend to each other in mixed-scale windows, which has two advantages: i) our mixed-scale windows are naturally overlapped with each other so that we do not have the misalignment problem as discussed in **Pillar-in-Pillar** and can further save the additional shift window operation which is commonly required by window-based transformers. ii) Without the need for multiple voxelizations, our model is more flexible and efficient and can be plugged into any single resolution layer to obtain multi-scale features.

---

> ### Author Response · Authors · 2022-08-02
> **Official Responses of Reviewer aXhD - Second Part**
>
> **R2-Q4: More experiments of mixed-window.**
> **R2-A4:** As shown below, we conduct a series of ablative experiments by increasing or decreasing the number of key windows and attention head groups allocated to each window. All models are trained for 12 epochs on 20\% Waymo data. We measure the latency on Tesla-V100 GPU and Unbuntu-16.04, Python 3.7, Cuda-10.2, and Pytorch-1.8. Compared with two windows (M1), more windows (M2, M3, M4) do not improve performance but increase the runtime latency. In our Scale-aware Head Attention, we distribute all feature channels and attention heads among the key windows to make different head groups in charge of a particular scale. Thus too scattered key windows will lead to fewer channels for each head group, which can weaken the representation ability of the single-scale features. Meanwhile, a single window (M5, M6) lacks multi-scale information and leads to low accuracy. Moreover, (M2, M3, and M4) also demonstrate that allocating more heads to the oversized windows significantly affects the accuracy of small objects. These experiments prove that our current combination (M1) can bring superiority of the mixed-scale window into full play and make a trade-off between speed and accuracy.
>
> | Strategy | Comb      | Num Heads | Veh / Ped / Cyc     | Lat (ms) |
> |----------|-----------|-----------|:-------------------:|:--------:|
> | M1*      | (3,7)     | (4,4)     | $72.37/75.99/67.90$ | 121      |
> | M2       | (3,5,7)   | (2,3,3)   | $71.94/75.76/67.07$ | 137      |
> | M3       | (3,5,7)   | (2,2,4)   | $72.17/75.45/67.01$ | 139      |
> | M4       | (3,5,7,9) | (2,2,2,2) | $72.04/75.38/66.53$ | 157      |
> | M5       | (3)       | (8)       | $71.46/74.39/65.55$ | 98       |
> | M6       | (7)       | (8)       | $72.06/75.20/66.33$ | 112      |
>
> **R2-Q5: More experiments of sampling strategy.**
> **R2-A5:** We conduct more experiments by adopting different sampling strategies for queries and keys. All models are trained for 12 epochs on 20\% Waymo data. As shown below, compared with CBS(query)+FPS(key), full adoption of FPS brings high latency and incurs a slight performance penalty on vehicles and pedestrians. Full adoption of CBS leads to faster speed and lower accuracy. In contrast, the CBS(query)+FPS(key) combination finds a sweet spot between latency and accuracy, where CBS significantly reduces the computation cost for query sampling and avoids the information loss caused by uncertain voxel updating. At the same time, FPS has a more vital ability to retain geometric structure.
>
> | Strategy |   Veh / Ped / Cyc   | Lat (ms) |
> |----------|:-------------------:|:--------:|
> | CBS+FPS* | $72.37/75.99/67.90$ | 121      |
> | FPS+FPS  | $72.02/75.57/67.91$ | 143      |
> | CBS+CBS  | $71.83/75.28/67.25$ | 107      |
> | FPS+CBS  | $71.88/75.04/67.33$ | 118      |

---

> ### Author Response · Authors · 2022-08-02
> **Official Responses of Reviewer aXhD - First Part**
>
> We sincerely thank the reviewer for providing thoughtful review and positive feedback. Below are our responses to the questions and suggestions raised by the reviewer.
>
> **R2-Q1: The motivation is a little inaccurate.**
> **R2-A1:** Thank you for your insightful comments. Your opinion on the multi-scale is very professional. In 3D detection, a large receptive field is essential to accurately regress various objects and recognize far-away sparse objects leveraging long-range contextual information. Meanwhile, fine-grained geometric details are crucial to improving objects' localization accuracy and distinguishing foreground objects from great false positives.
>
> However, we found that directly applying the attention mechanism in a large window will blur out the fine-grained details, while restricting attention range in a small window leads to an insufficient receptive field. Inspired by this phenomenon, we propose the mixed-scale window design, which can keep finer granularity by applying self-attention in densely sampled nearby voxels within small windows and simultaneously expand the receptive field by attending to roughly collected distant voxels. This way, the problem of the sparsity and irregularity of LiDAR points distribution and the scale variance of different instances can be alleviated.
>
> 1) As for the challenge of the ununiform distribution of point clouds, MsSVT can gather long-range contextual information to accurately detect the far-away objects in the absence of fine-grained information (e.g., for the vehicle only contain 1-5 points, we can recognize it and correctly infer the box from its surroundings such as other vehicles and the road), as shown in appendix (E). Meanwhile, our single-stage detector outperforms the SOTA two-stage detectors PV-RCNN++[34] without refining one-stage boxes by complicated RoI module as in the latter, demonstrating our strong ability to utilize fine-grained details to improve the localization accuracy.
>
> 2) Though not as apparent as in the 2D images, the absolute scale variance also challenges 3D detection (e.g., the large vehicle can be 15m-30m long while pedestrian only ranges from 0.5m-1m in the Waymo Open Dataset). Previous work such as SST[7] achieves impressive results on small objects by avoiding down-sampling to maintain fine-grained features. In contrast, our method achieves strong results on small and large objects. Notably, we achieve remarkable improvements by **+4.73, +1.34** for vehicle and pedestrian in terms of LEVEL2 mAP compared with SST[7], which uses single scale window, demonstrating the significant superiority of the mixed-scale window design.
> In addition, we have revised the sentence in the abstract to avoid misunderstanding.
>
> **R2-Q2: Additional results on KITTI test set.**
> **R2-A2:** Due to limited space, we do not report the results of the KITTI test set in the main paper. We submit our results to KITTI's official test server and provide the results below. Our method beats the superior PV-RCNN++ by $0.6$ mAP. The results on KITTI demonstrate that our MsSVT has excellent generalizability to adapt to different datasets. We will add the results in the final version.
>
> | Method\\mAP@Car  | Easy  | Mod   | Hard  |
> |-----------------|:-----:|:-----:|:-----:|
> | SECOND          | 84.65 | 75.96 | 68.71 |
> | PV-RCNN         | **90.25** | 81.43 | 76.82 |
> | PV-RCNN++       | 90.14 | 81.88 | 77.15 |
> | MsSVT-SS (ours) | 90.04 | **82.10** | **78.25** |
>
> **R2-Q3: Better to put two sections about sampling together, the paper needs polishing and fixing some minor typos.**
> **R2-A3:** Thank you for your suggestion. We agree that putting sections of Balanced Multi-window Sampling and Chessboard sampling together can make it more transparent and easy to understand. We have updated the methodology, carefully polished our writing, and fixed the typos in the attached revision.

---

### Official Review · Reviewer_QRYf · 2022-07-11

**Rating:** 4
**Confidence:** 5
**Soundness:** 3 good
**Presentation:** 2 fair
**Contribution:** 2 fair

**Summary:**

This paper presents a novel Mixed-scale Sparse Voxel Transformer to capture both types of information simultaneously by the divide-and-conquer philosophy. Experiments on waymo show promising results.

**Questions:**

Refer to part Strengths And Weaknesses.

**Limitations:**

Refer to part Strengths And Weaknesses.

**Strengths And Weaknesses:**

1. The general idea of this article is to extract features through multi-scale transformers in point cloud voxels, and then perform regression through 2D CNN. First of all, the overall process and idea are exactly the same as PointPillar. Second, this multi-scale idea is the same as Voxel-FPN[2], but the processing backbone is replaced by a transformer. So from this point of view, the author needs to think carefully about the novelty of this article.

[1] Lang A H ,  Vora S ,  Caesar H , et al. PointPillars: Fast Encoders for Object Detection From Point Clouds[C]// 2019 IEEE/CVF Conference on Computer Vision and Pattern Recognition (CVPR). IEEE, 2019.

[2] Wang B ,  An J ,  Cao J . Voxel-FPN: multi-scale voxel feature aggregation in 3D object detection from point clouds[J].  2019.

2. This article is poorly written and needs careful polishing, such as line 5, line 32.

---

> ### Author Response · Authors · 2022-08-02
> **Official Responses of Reviewer QRYf - Second Part**
>
> **R1-Q2: The multi-scale idea is the same as Voxel-FPN, but the processing backbone is replaced by a transformer.**
> **R1-A2:** Thanks. **Voxel-FPN** [15] is a valuable work in 3D detection and has been involved in the attached revision. Voxel-FPN adopts a multi-scale voxelization strategy to rasterize the input point clouds into voxels with different voxel sizes and utilizes an FPN network to aggregate the multi-resolution feature maps. **Pillar-in-Pillar** [40] further solves the misalignment problem in multi-scale voxelization by leveraging a center-aligned voxelization strategy with overlapped sub-voxel partition.
> Our **MsSVT** is very different from these multi-scale voxelization methods.
>
> 1) **We gather mixed-scale information from different windows rather than apply multiple voxelizations.** First of all, our mixed-scale windows are different from the multi-resolution voxels in two aspects: i) The basic unit of our windows is still a high-resolution voxel, while Voxel-FPN pools all the information in a significant voxel, in which case the local structure will be blurred irreversibly. For example, the low-resolution voxelization will convert an object into a solid cube, while our large window roughly collects high-resolution voxels on its surface and thus can still retain the local structure thanks to the well-designed sampling strategy. ii) Our windows are much larger than the multi-resolution voxels (e.g., we have two window sizes (1.2m, 2.8m) while Voxel-FPN adopts three voxel sizes (0.16m, 0.32m, 0.64m)). This way, our MsSVT can obtain sufficient receptive benefit from the large window more directly and efficiently.
>
> 2) **MsSVT can capture mixed-scale information in a single layer while Voxel-FPN relies on a complicated architecture.** Previous multi-scale voxelization methods must first adopt multiple voxelizations, then stack 3D CNNs for different resolutions to extract multi-scale feature maps, and finally utilize the FPN network to fuse the pyramid feature maps and obtain the multi-scale features. Technically, the multiple voxelizations will bring disastrous computational costs to the sparse voxel operation. While our **MsSVT** only voxelizes the point clouds at once at the beginning, then maintains the resolution without down-sampling as SST [7]. We can capture multi-scale information in a single layer by making voxels learn to attend to each other in mixed-scale windows, which has two advantages: i) our mixed-scale windows are naturally overlapped with each other so that we do not have the misalignment problem as discussed in **Pillar-in-Pillar** and can further save the additional shift window operation which is commonly required by window-based transformers. ii) Without the need for multiple voxelizations, our model is more flexible and efficient and can be plugged into any single resolution layer to obtain multi-scale features.
>
> **R1-Q3: This article is poorly written and needs careful polishing, such as line 5, line 32.**
> **R1-A3:** Thanks. We have carefully polished our writing and fixed the grammatical errors in the attached revision.

---

> > ### Author Response · Authors · 2022-08-08
> > **Responses of Reviewer QRYf**
> >
> > Dear reviewer, we have answered your questions in the author response and also uploaded a revised manuscript by following your suggestions for paper writing. We hope that we have addressed all your concerns. Do you have any further assessment (or concerns) of our work? Thanks for your kind consideration.

---

> ### Author Response · Authors · 2022-08-02
> **Official Responses of Reviewer QRYf - First Part**
>
> We sincerely thank the reviewer for providing the thoughtful review. Below are our responses.
>
> **R1-Q1: The overall process and idea are exactly the same as PointPillar.**
> **R1-A1:** In outdoor 3D detection, Pointpillar [16] and SECOND [45] represent two mainstream paradigms, and almost all the well-known methods adopted these two basic paradigms and improved some modules in these two frameworks, such as CenterPoint [51] (detection head), SST [7] (3D backbone), VoTR [22] (3D backbone) and PillarNet [31] (2D encoder). Pointpillar directly converts the point cloud into 2D pillars and only utilizes 2D CNNs to build the network, which is more efficient but yields unsatisfied performance since all the operations are performed in 2D and the 3D structure information can not be well extracted. SECOND rasterizes the points into 3D voxels and extracts features through a 3D convolution backbone, which maintains the 3D structure and thus achieves better performance.
>
> However, the 3D CNNs in backbone can not provide sufficient receptive field, which is crucial for large-scene 3D detection. Moreover, the shared weight attribute of CNN makes it unable to model the local regions adaptively, while the sparse and irregular points change violently in different areas and need to be considered dynamically. Currently, the window-based transformer, which has revolutionized 2D image processing, shows a powerful ability to capture long-range information and is adaptive to the input, providing a potential solution for the above problems.
>
> In our work, we make great efforts to design a novel 3D transformer backbone suitable for point cloud 3D detection while leveraging the incredible power of window-based transformers. To make a fair comparison, we adopt the SECOND paradigm and replace its pivotal 3D CNN backbone with our Mixed-scale Sparse Voxel Transformer (MsSVT), which achieves significant improvements, **+6.22, +15.02, +16.62** for the vehicle, pedestrian, and cyclist respectively on the challenging Waymo Open Dataset. Moreover, we overcome many difficulties when introducing window-based transformers into 3D voxels through our novel architecture design and technical contribution.
>
> 1) **We conquer the ununiform sampling bias problem through mixed-scale window design and balanced multi-window sampling strategy.** Current window-based transformers achieve great success by applying an attention mechanism in a single-scale large window.  However, the memory occupation cubically grows when migrating to 3D voxel space. Thus we have to sample a certain number of voxels in the window to reduce computation costs. Unfortunately, this simple sampling strategy will miss significant nearby voxels. It tends to sample more distant voxels when the object has few voxels (e.g., far-away objects only contain 1-5 voxels more likely to be irreversibly dropped). Meanwhile, it cannot collect enough distant voxels from enlarging the receptive field when it comes to dense areas and will bias these dense voxels (e.g., dense background area). To solve this problem, we divide the local region into mixed-scale windows and adopt balanced multi-window sampling strategy: collecting all the voxels in the small window to avoid loss of fine-grained information while using FPS to obtain a uniformly sampled distant voxel set in the large window, as shown in Figure 1(c).
>
> 2) **We leverage the scale-aware head attention to achieve effective mixed-scale information aggregation.** It is not trivial to simultaneously capture the fine-grained details and long-range contextual information through the transformer. In our early exploration, putting all the nearby and distant voxels together and making the network learn to attend them alone do not bring improvements because of the diverse distribution of voxels from mixed-scale windows. In our work, we present the Scale-aware Head Attention to explicitly make different attention heads in charge of attending to information within a particular range (window). In this way, we solve the problem of ununiform distribution and significantly boost the performance. In addition, we also design a new Scale-aware Relative Position Encoding to enhance position information in different attention heads.
>
> 3) **We significantly reduce the memory and computation cost by chessboard sampling and sparse implementation.** In general, the unacceptable memory and computation cost caused by the cubically growing queries and keys when applying attention in 3D voxels significantly limit the development of the 3D transformer backbone. In our work, we present the balanced multi-window sampling strategy, which can reduce the number of keys. Besides, we also present a novel chessboard sampling to reduce the number of queries without loss of information, which significantly reduces the memory and improves the speed, as shown in Table 5. In addition, we implement all the operations in MsSVT sparsely and accelerate them in parallel on GPU.

---

### Meta-Review · Area_Chair_QaMt · 2022-08-25

**Recommendation:** Accept
**Confidence:** Certain

**Metareview:**

After the rebuttal and discussion two reviewers are positive, one remains negative. The reviewers liked the overall approach, the writing, and the core experimental results. Some reviewers asked for additional broader experiments and comparisons, which the authors were able to provide. The main concern of reviewer QRYf is the close relation to PointPillar's, which the authors are able to clarify sufficiently in their rebuttal.
There is thus sufficient evidence to accept this submission.

**Award:**

No

---

### Decision · Program_Chairs · 2022-09-14

Accept